# A New Family of Transcriptional Regulators Activating Biosynthetic Gene Clusters for Secondary Metabolites

**DOI:** 10.3390/ijms23052455

**Published:** 2022-02-23

**Authors:** Renata Novakova, Erik Mingyar, Lubomira Feckova, Dagmar Homerova, Dominika Csolleiova, Bronislava Rezuchova, Beatrica Sevcikova, Rachel Javorova, Jan Kormanec

**Affiliations:** Institute of Molecular Biology, Slovak Academy of Sciences, 845 51 Bratislava, Slovak; renata.novakova@savba.sk (R.N.); erik.mingyar@gmail.com (E.M.); lubomira.feckova@savba.sk (L.F.); dagmar.homerova@savba.sk (D.H.); dominika.csolleiova@savba.sk (D.C.); bronislava.rezuchova@savba.sk (B.R.); beatrica.sevcikova@savba.sk (B.S.); rachel.javorova@savba.sk (R.J.)

**Keywords:** antibiotics, auricin, polyketide, regulation, secondary metabolite, *Streptomyces*

## Abstract

We previously identified the *aur1* biosynthetic gene cluster (BGC) in *Streptomyces*
*lavendulae* subsp. *lavendulae* CCM 3239 (formerly *Streptomyces*
*aureofaciens* CCM 3239), which is responsible for the production of the unusual angucycline-like antibiotic auricin. Auricin is produced in a narrow interval of the growth phase after entering the stationary phase, after which it is degraded due to its instability at the high pH values reached after the production phase. The complex regulation of auricin BGC is responsible for this specific production by several regulators, including the key activator Aur1P, which belongs to the family of atypical response regulators. The *aur1P* gene forms an operon with the downstream *aur1O* gene, which encodes an unknown protein without any conserved domain. Homologous *aur1O* genes have been found in several BGCs, which are mainly responsible for the production of angucycline antibiotics. Deletion of the *aur1O* gene led to a dramatic reduction in auricin production. Transcription from the previously characterized Aur1P-dependent biosynthetic *aur1Ap* promoter was similarly reduced in the *S. lavendulae*
*aur1O* mutant strain. The *aur1O*-specific coactivation of the *aur1Ap* promoter was demonstrated in a heterologous system using a luciferase reporter gene. In addition, the interaction between Aur1O and Aur1P has been demonstrated by a bacterial two-hybrid system. These results suggest that Aur1O is a specific coactivator of this key auricin-specific positive regulator Aur1P. Bioinformatics analysis of Aur1O and its homologues in other BGCs revealed that they represent a new family of transcriptional coactivators involved in the regulation of secondary metabolite biosynthesis. However, they are divided into two distinct sequence-specific subclasses, each of which is likely to interact with a different family of positive regulators.

## 1. Introduction

Gram-positive soil bacteria of the genus *Streptomyces* (class *Actinobacteria*, order *Actinomycetales*, family *Streptomycetaceae*) are characterized by their ability to produce a wide range of bioactive secondary metabolites, including many known antibiotics. They undergo an exceptional process of morphological differentiation, initiated by germination of spores to form a network of branched multinucleoid hyphae (a so-called substrate mycelium). In response to various signals, it differentiates into white air-grown hyphae (a so-called aerial mycelium), which eventually undergo septation into unigenomic spore chains. Secondary metabolite production is coordinated with morphological differentiation and often occurs at the time of aerial mycelium formation or at the end of the exponential growth phase during growth in a liquid medium [1,2].

The genes responsible for the biosynthesis of antibiotics and other secondary metabolites are typically grouped into so-called biosynthetic gene clusters (BGCs) together with genes encoding pathway-specific or cluster-situated regulatory proteins. These regulatory genes located in BGCs are controlled by global regulators that integrate physiological and environmental signals to control the production of multiple secondary metabolites and have pleiotropic roles in stress response and morphological differentiation. In general, the production of secondary metabolites in streptomycetes is tightly and elaborately regulated by pyramidal regulatory cascades, including global regulators with the primary role of sensing various intracellular and extracellular signals and pathway-specific or cluster-situated regulatory proteins that are specifically regulated by these global regulators. In addition, feedback control of global regulators by pathway-specific regulators has been characterized in some regulatory cascades. These networks determine the production of antibiotic under specific culture condition. A number of global regulators have been characterized in *Streptomyces* (e.g., AbsA2, AdpA, AfsQ1, AtrA, DasR, GlnR, PhoP/PhoR, WblA) [3,4,5,6]. Among these global regulatory systems, those that use small diffuse hormone-like signaling molecules, including γ-butyrolactone (GBL) autoregulator, furans, γ-butenolides, PI factor, and N-methylphenylalanyl-dehydrobutyrine diketopiperazine have been most characterized. The best characterized prototype GBL system consists of a GBL synthase, responsible for GBL biosynthesis, and a related GBL receptor, which belong to the TetR family. In the absence of GBL, it binds a specific DNA sequence in front of its target genes (including pathway-specific regulators among others), thereby suppressing their transcription, usually during the exponential phase. Binding of a specific GBL autoregulator to the GBL receptor prevents this binding, thereby allowing induction target gene expression. Through this mechanism, GBLs regulate and coordinate production of secondary metabolites and, in some cases, also affect morphological differentiation [4,7].

Pathway-specific transcriptional regulators have been classified into different families based on sequence or structure similarities. The most common and best studied is the *Streptomyces*
antibiotic regulatory protein (SARP) family. These members are activators characterized by an N-terminal winged helix-turn-helix (HTH) DNA-binding domain of the OmpR-type. The first prototype of this family studied was ActII-ORF4 for the aromatic polyketide actinorhodin in *Streptomyces coelicolor*. Another widespread family is LAL (large ATP-binding regulators of the LuxR family), whose genes encode large transcriptional activators and are located predominantly in the BGC for the type I PKSs, such as *pikD* for pikromycin in *Streptomyces venezuelae*. The transcriptional activator StrR, which controls the biosynthesis of aminoglycoside antibiotic streptomycin in *S. griseus*, is a prototype of another family designated by this regulator. It represents the first complete regulatory pathway leading to BGC activation. Interestingly, unusually, it belongs to the ParB-Spo0J family of DNA segregation proteins. Genes of this family are found in most glycopeptide BGCs. Another common family includes atypical response regulators (ARRs), which show homology to the OmpR family of response regulators of bacterial two-component signal transducing systems. However, they lack the residues necessary for phosphorylation as well as the accompanying sensor histidine kinase genes. They are characterized by an N-terminal receiver domain and a winged HTH C-terminal domain. The most studied member of this ARR family is the activator JadR1 for the angucycline antibiotic jadomycin in *S. venezuelae*. Interestingly, unlike classical response regulators, JadR1 is feedback-regulated with jadomycin. Genes for most other members of this ARR family have been found in angycycline BGCs and encoded transcriptional activators (e.g., LanI, LndI, Aur1P, GcnR, ChaI, SchA25, SimReg1), which are essential for the biosynthesis of the corresponding antibiotics in other *Streptomyces* spp. In addition, a similar feedback regulatory mechanism through end-product binding was found for two other homologues, Aur1P and SimgReg1. In addition to transcriptional activators, many BGCs are often negatively regulated by TetR family transcriptional regulators. Members of this family are widespread in various bacteria and act mainly as repressors to regulate genes for biosynthetic enzymes for antibiotics, drug-efflux pumps and other proteins. The structure of the TetR-family proteins consists of two domains: the N-terminal HTH DNA-binding domain and the C-terminal regulatory domain which binds the ligand, leading to the loss of its DNA-binding activity and the subsequent activation of transcription of the target gene [1,3,4,5,6,8].

Initial sequence analysis of *aur1* BGC in *Streptomyces aureofaciens* CCM 3239 revealed 15 open reading frames that showed high similarity to several angucycline type II polyketide synthase (PKS) BGCs, which was responsible for the antibiotic auricin [9]. We later found that the cluster is not located in the chromosome, but on the large linear plasmid pSA3239 [10]. However, a recent genomic sequence of this strain (GenBank Acc. No. CP024985) revealed that it was incorrectly administrated by the Czech Collection of Microorganisms (CCM) and is, in fact, *Streptomyces lavendulae* subsp. *lavendulae* CCM 3239 [11]. The complete sequence of auricin BGC, including its contiguous regions, revealed its unusual organization. It consists of a central region (*aur1A-aur1N*), which contains polyketide aglycone biosynthetic genes that were very similar to the angucycline biosynthetic genes. However, a number of auricin-specific tailoring biosynthetic genes were scattered up to 30 kb from this central region. In addition, auricin BGC contains an unusually large number of genes encoding regulatory proteins of different families (Figure 1) [8,12,13].

Interestingly, in the liquid medium, auricin is transiently produced during a narrow interval after entering the stationary phase, after which it degrades to inactive metabolites due to its instability at high pH values, which are reached later in the stationary phase [14]. Structural analysis revealed that auricin has interesting structural features that set it apart from all other known angucyclines. It is modified with d-forosamine and contains a unique aglycone similar to griseusin-like spiroketal pyranonaphthoquinones [15]. 

This unusual production of auricin is due to its strict and complex regulation of its biosynthesis, which involves both forward and feedback control by auricin intermediates upon interaction with several pathway-specific regulators [8,14]. Some of them have already been characterized. Aur1P, which belongs to the ARR family, is a key positive regulator. It activates the expression of the core auricin biosynthetic genes (*aur1A-aur1U*) (Figure 1) from the *aur1Ap* promoter [16]. This activation is abolished by some auricin intermediates to create feedback control of auricin production [14]. The Aur1R repressor belonging to the TetR family suppresses the expression of the *aur1P* gene by direct binding of its *aur1Pp* promoter. Some auricin intermediates break this bond [17]. Two other characterized positive regulators, Aur1PR3 and Aur1PR4, belong to the SARP family. They play a role in the regulation of auricin-tailoring biosynthetic genes. In addition, the expression of their genes is under different control with Aur1R and Aur1P [12,18,19]. At the global level, the SagA/SagR GBL autoregulator-receptor system controls the expression of pathway-specific *aur1P* and *aur1R* regulatory genes. In addition, this GBL system is regulated by a feedback mechanism with Aur1R [20]. Auricin regulation is summarized in Figure 1. 

The aim of the present study was to characterize the role of another putative *aur1O* regulatory gene, located just behind the key auricin-specific regulatory gene *aur1P*, in the regulation of auricin. For this purpose, the *aur1O* gene was inactivated and auricin production was examined in the resulting mutant. Gene expression analysis using S1-nuclease mapping was used to characterize the expression of the *aur1Ap* biosynthetic promoter and its dependence upon *aur1O*. The putative role of the Aur1O as a co-activator of Aur1P was investigate in a heterologous system using the luciferase reporter gene. In addition, the interaction between Aur1O and Aur1P were investigated by a bacterial two-hybrid system. The results showed that Aur1O binds Aur1P and is a specific coactivator of this key auricin-specific positive regulator. Bioinformatics analysis of Aur1O in other BGCs revealed that Aur1O and its homologues represent a new family of transcriptional coactivators involved in the regulation of secondary metabolite biosynthesis, which have two different interacting positive regulators as partners. Their genes are present predominantly in *Streptomyces* spp. and are found mainly in angucycline BGCs. 

## 2. Results and Discussion

### 2.1. Characterization of the aur1O Gene in Auricin Biosynthesis

We have previously identified the auricin-specific activator Aur1P required for auricin production [16]. Its gene is followed by the *aur1O* gene, and both genes form an operon driven by the *aur1Pp* promoter (Figure 1), because no other promoter has been identified before the *aur1O* gene. To investigate the role of the *aur1O* gene in auricin biosynthesis, the gene was inactivated in *S. lavendulae* subsp. *lavendulae* CCM 3239 using a PCR targeting system designed to disrupt *Streptomyces* genes [21]. The apramycin-resistance (AprR) marker gene was used to replace the entire *aur1O* gene. The strategy resulted in four independent mutant *S. lavendulae* ∆*aur1O::AprR* clones, which were verified by Southern blot hybridization analysis (Figure 2). All clones had similar phenotypes and the deletion of *aur1O* did not affect growth and differentiation (data not shown). One representative mutant strain was selected for further study.

Auricin production in the prepared *S. lavendulae* ∆*aur1O::AprR* mutant strain was analyzed as described previously [12]. The *S. lavendulae* ∆*aur1O::AprR* mutant strain, as well as wild-type as a control, were cultured in liquid Bennet medium, and ethyl acetate extracts were prepared from several time points. Extracts were analyzed by TLC followed by biochromatography with *Bacillus subtilis* as the test strain. In parallel, the extracts were analyzed by HPLC. The deletion of *aur1O* had a dramatic effect on auricin production. The yellow spots and inhibition zones corresponding to auricin (Rf = 0.13) in the mutant were smaller than in the wild-type strain at all time points examined (Figure 3a,b). HPLC analysis similarly showed a significant reduction in the peak corresponding to auricin in the mutant extract, compared to the wild type (Figure 3c). Determination of the auricin level revealed a 4.6-fold decrease in the *aur1O* mutant, compared to the wild-type strain. To confirm that this reduced peak in the *aur1O* mutant corresponds to auricin, this peak was isolated and high resolution ESI MS analysis revealed a molecular ion [M + H]^+^ at *m*/*z* 542.2014. This value corresponded exactly to auricin [14]. To verify that this phenotype was due to deletion of *aur1O*, the mutant strain *S. lavendulae* ∆*aur1O::AprR* was complemented in trans by transformation with plasmid pAPHII15-aur1O, which contained the *aur1O* gene under the control of the *ermEp** promoter [23]. The level of auricin in the complemented strain was similar to that of wild-type *S. lavendulae* subsp. *lavendulae* CCM 3239 (data not shown). It confirmed that the decrease of auricin in the mutant strain *S. lavendulae* ∆*aur1O::AprR* is indeed due to the deletion of *aur1O*. These results suggest that the *aur1O* gene has no role in auricin biosynthesis but plays an important role in the positive regulation of auricin biosynthesis. 

### 2.2. Transcriptional Analysis of the aur1Ap Promoter in the aur1O Mutant

We have previously characterized the *aur1Ap* promoter (Figure 1), which directs the expression of the *aur1A-aur1U* operon containing the auricin biosynthetic genes of its BGC. During growth in liquid Bennet medium, the *aur1Ap* promoter in not active in the exponential phase and is induced upon entry into the stationary phase. In addition, the promoter is induced at the beginning of aerial mycelium formation during growth and differentiation on solid Bennet medium. The activity of the promoter was dependent upon the auricin-specific transcriptional activator Aur1P, which binds directly to the promoter to activate its transcription [8,9,16]. To investigate whether the deletion of *aur1O* affects the transcription of the *aur1Ap* promoter, S1-nuclease mapping was performed using RNA isolated from wild-type *S. lavendulae* subsp. *lavendulae* CCM 3239 and mutant *S. lavendulae* ∆*aur1O::AprR* strains at different growth stages in liquid Bennet medium. A single RNA-protected fragment was identified with the level of *aur1Ap* mRNA induced after entering the stationary phase of growth in the wild-type strain, similar to previous reports [9,16,18]. However, the level of *aur1Ap* mRNA from all time points was dramatically reduced in the *aur1O* mutant. No RNA-protected fragment was identified by tRNA control (Figure 4a). As an internal control of RNA quality, S1-nuclease mapping was performed with the same RNA samples using a probe for the *hrdBp2* promoter, which is continuously expressed during growth [24]. RNA-protected fragments of similar intensities corresponding to the *hrdBp2* promoter were identified in all RNA samples (Figure 4b). These results suggest that the *aur1Ap* promoter is dependent on the *aur1O* gene. Decreased transcription of this first *aur1Ap* biosynthetic promoter is probably responsible for reduced auricin production in the *aur1O* mutant.

### 2.3. Activation of the aur1Ap Promoter by aur1P and aur1O in the Heterologous System

Because both *aur1P* and *aur1O* genes form an operon, it is possible that both are involved in activating the *aur1Ap* promoter. Aur1P contains the HTH DNA binding domain at the N-terminus and directly binds to and activates the *aur1Ap* promoter [16]. However, Aur1O does not contain any DNA-binding motif or other conserved domain. Therefore, it may affect the binding of Aur1P to the *aur1Ap* promoter. We used a heterologous system to test the effect of Aur1P and Aur1O on the transcription of the *aur1Ap* promoter. A DNA fragment containing the *aur1Ap* promoter, including its Aur1P-binding site [16], was cloned into the luciferase reporter vector pMU1s*. This PhiBT1 phage-based integration plasmid contains the synthetic *luxCDABE* operon and is able to integrate in a single copy into the *Streptomyces* chromosome [26]. The resulting recombinant plasmid pMU1s-aur1Ap (Figure 5a) was conjugated to the heterologous host strain *Streptomyces coelicolor* M1146 [27] and the luminescence of eight independent AprR clones was determined during differentiation on solid Bennet medium. The luminescence of all clones was at the background level of approximately 200 relative luminescence units (RLU), indicating that the *aur1Ap* promoter is not active in the heterologous *S. coelicolor* M1146 strain (Figure 5b). The *aur1P* gene, including its *aur1Pp* promoter, was inserted into pMU1s-aur1Ap, resulting in pMU1s-aur1PAp (Figure 5a), which was similarly conjugated to *S. coelicolor* M1146, and the luminescence of eight independent AprR clones was determined. The activity of the *aur1Ap* promoter in this construct increased during growth and its maximum (on average 118,030 RLU) coincided with the onset of aerial mycelium formation (Figure 5b). In general, the production of secondary metabolites coincides with the initiation of aerial mycelium formation, and the *aur1Ap* promoter was similarly induced in this stage in original *S. lavendulae* subsp. *lavendulae* CCM 3239 strain [9]. This result confirmed the activation of the *aur1Ap* promoter by the auricin-specific transcriptional activator Aur1P also in the heterologous *S. coelicolor* M1146 strain. In contrast to *aur1Ap*, the *aur1Pp* promoter is highly active in the *S. coelicolor* M1146 heterologous system with a similar time course as the activated *aur1Ap* promoter, with increasing activity from onset to maximum at 64 h (on average 65,050 RLU) (data not shown). In *S. lavendulae* subsp. *lavendulae* CCM 3239, both negative regulators, SagR (GBL receptor) and Aur1R (pathway-specific pseudo GBL receptor), bind the *aur1Pp* promoter to downregulate its activity prior to auricin production (Figure 1). The heterologous *S. coelicolor* M1146 strain contains the homologue of SagR (ScbR, SCO6265), but does not contain the homologue of Aur1R (ScbR2, SCO6286), because this strain contains a deletion of the *cpk* BGC from SCO6270 to SCO6286 [27]. The high increasing activity of the *aur1Pp* promoter in this *S. coelicolor* M1146 heterologous system suggests that the homologous GBL receptor ScbR did not significantly affect the activity of the *aur1Pp* promoter and consequently the level of Aur1P under these conditions. The *aur1P-aur1O* operon, including the *aur1Pp* promoter, was similarly inserted into pMU1s-aur1Ap, resulting in pMU1s-aur1POAp (Figure 5a). This construct was similarly conjugated to *S. coelicolor* M1146 and the luminescence of eight independent clones was determined. The activity of the *aur1Ap* promoter in this construct was 3.22-fold higher (on average 380,814 RLU) and its maximum similarly coincided with the onset of aerial mycelium formation (Figure 5b). This result showed a positive effect of Aur1O on the activation of the *aur1Ap* promoter by Aur1P.

### 2.4. aur1O Interacts with aur1P

The results of the above experiments suggest that Aur1O together with Aur1P activate the *aur1Ap* promoter. Because Aur1O does not contain a DNA binding domain, it can exert its activation function on the *aur1Ap* promoter through interaction with Aur1P. A bacterial two-hybrid (BACTH) system was used to study the interaction between these proteins. This system is based on functional complementation between *Bordetella pertussis* adenylate cyclase fragments T18 and T25, expressed separately from the two compatible plasmids pKT25 and pUT18C [28]. The full-length *aur1P* and *aur1O* genes were fused as C-terminal fusions in plasmids pKT25 and pUT18C, resulting in pKT25-aur1P, pKT25-aur1O, pUT18C-aur1P, and pUT18C-aur1O. These constructs (including the pKT25+pUT18C negative control) were co-transformed into *E. coli* BTH101 and screened on LB medium with isopropyl-β-d-thiogalactopyranoside (IPTG) and X-Gal. The negative control correctly created white colonies. Both combinations showed a LacZ+ phenotype and formed blue colonies (Figure 6a). Measurement of β-galactosidase activity in sets of three independent experiments for each combination of plasmids was used to quantify protein interactions. The level of β-galactosidase activity was similar in both combinations and significantly higher than in the negative control (Figure 6b). The results showed that Aur1O interacts with Aur1P. Thus, the role of Aur1O may be to coactivate the auricin-specific activator Aur1P.

### 2.5. Presence of aur1O Homologues in Other BGCs

The *aur1O* gene product showed high similarity to various proteins of unknown function (Figure 7a), whose genes were found in several antibiotic BGCs, especially in *Streptomyces* spp. (Figure 8). This suggests that these homologues could play a similar role in their corresponding BGCs. All, including Aur1O, do not contain any conserved domain. Most BGCs containing the gene encoding the *aur1O* homologue belong to the type II PKS for angucycline aromatic polyketides, including the *aur1* for auricin; *cha* is responsible for the production of chattamycin in *S. chattanoogensis* [29], *saq* for saquayamycin in *Micromonospora* sp. Tu6368 [30], *pga* for gaudimycin in *Streptomyces* sp. PGA64 [31], *sch* for angycyclines Sch 47554 and Sch 47555 in *Streptomyces* sp. SCC-2136 [32], *gcn* for grincamycin in *S. lusitanus* SCSIO LR32 [33], *ovm* for oviedomycin in *S. antibioticus* ATCC 11891 [34], and *lac* for unknown angycycline in *S. lavendulae* FRI-5 [35]. However, some of the BGCs are responsible for the biosynthesis of secondary metabolites belonging to other structural types; *med* belongs to type II PKS for the aromatic polyketide medermycin in *Streptomyces* sp. AM-7161, which belongs to the class of pyranonaphthoquinones [36], *tyl* belongs to type I PKS responsible for the macrolide polyketide tylosin in *S. fradiae* [37], two BGCs, *epa* and *phz*, are responsible for the biosynthesis of phenazines in *Kitasatospora* sp. HKI 714 and *S. tendae* Tue1028 [38,39], and *nap* is responsible for napyradomycin in *S. tendae* Tue1028, which belongs to a specific family of chlorinated meroterpenoid dyhydroquinones [40]. 

There are only two reports in which the function of the Aur1O homologue was partially characterized. In the first, the *tylU* gene was identified in type I PKS BGC for tylosin in *S. fradiae*. Its product has been shown to be important for tylosin production because deletion of *tylU* resulted in reduced tylosin yield of approximately 80%. Tylosin production is regulated by several regulators, and TylU is thought to play a regulatory role in the production of tylosin as a helper protein for the tylosin positive SARP-family regulator TylS. TylU is thought to promote the binding of TylS to a promoter that directs the expression of another tylosin-specific activator gene *tylR*, whose product is directly involved in the activation of tylosin biosynthesis genes [37]. These results of the role of TylU in tylosin production are similar to our case, with the difference that the auricin-specific activator Aur1P does not belong to the SARP family, but to the ARR family [14,16,41]. In addition, both Aur1P and Aur1O proteins directly activate the *aur1Ap* biosynthetic promoter. In two other reports, the *ovmZ* gene was identified in the type II PKS BGC for the angucycline antibiotic oviedomycin. The *ovmZ* gene identified in the *S. antibioticus* ATCC 118911 has not been characterized [34]. However, in homologous oviedomycin BGC in *S. ansochromogenes*, its product was shown to be important for oviedomycin production because deletion of *ovmZ* led to loss of oviedomycin production. Interestingly, *ovmZ* was co-transcribed with the small downstream *ovmW* gene (Figure 8), which was similarly necessary for oviedomycin production. Like OwmZ, OvmW does not contain any conserved domain. However, unlike OvmZ, OwmW contains the HTH DNA binding domain of the HTH_17 family (pfam12728). Both genes have been defined as positive regulators necessary for oviedomycin BGC activation. They are interdependent and cooperate as partners. As in our case, expression of both *ovmZ*/*ovmW* genes activated the *ovmOp* biosynthetic promoter fused to the *gusA* reporter gene in the *S. coelicolor* M1146 heterologous system [42]. 

Interestingly, unlike *ovmZ*, no *ovmW* orthologue was identified downstream of the *aur1O* gene, even in the entire auricin BGC, and in several other BGCs, except *tyl*, *nap*, *epa*, and *phz* (Figure 8). Alignment of the amino acid sequences of Aur1O homologues revealed several conserved domains (Figure 7a). Interestingly, all Aur1O homologues whose genes are translationally coupled with the *ovmW* homologue (TylU, OvmZ, NapU1, Phz-orf13, EpaI) (Figure 8) are shorter and do not have a conserved C-terminal domain VI containing many aromatic and basic amino acids (Figure 7a). In addition, a phylogenetic tree containing Aur1O homologues (Figure 7b) shows that all homologues that likely cooperatively function with the OvmW homologue belong to a separate branch.

Interestingly, almost all other BGCs lacking the homologous *owmW* gene contain a gene encoding ARR similar to the auricin-specific Aur1P positive regulator (Appendix A). In two cases (*chaI*, *farR1*) [29,43], this gene is located near the *aur1O* homologue (Figure 7). In other cases, they are placed in a different position within the BGC; the homologous *pgaR1* gene is located approximately 4.5 kb downstream of the *pgaK* gene in *pga* [31], the *schA25* gene is located approximately 4.5 kb upstream of the *schP11* gene in *sch* [32], the *med-ORF30* gene is located approximately 4 kb downstream of the *med-ORF27* gene in *med* [36], and the *gcnR* gene is located approximately 27 kb downstream of the *gcnE* gene in *gcn* [33]. There is an exception in *saq* BGC, where the putative transcriptional activator gene *saqI*, located near *saqP* (Figure 8), belongs to the StrR family [30]. Both genes were located near the beginning of BGC, so it cannot be rule out that a gene homologous to *aur1P* is located in a more upstream region, outside the sequenced region. Assuming that both Aur1O and Aur1P homologues can interact as in our case, it is possible that the conserved C-terminal domain specific for these Aur1O homologues may be involved in interacting with these transcriptional regulators. In addition to the C-terminal conserved DNA-binding domain, all these ARRs are conserved in the N-terminal effector domain, which may be involved in this interaction (Appendix A).

All these results suggest that the Aur1O homologues represent a new family of transcriptional coregulators that play a role in activating their BGC in association with two different DNA binding regulators. Shorter Aur1O homologues (TylU, NapU1, EpaI, OvmZ, Phz-ORF13) may exert their positive function upon interaction with the product of small *ovmW* homologous gene, which contains a DNA binding domain, as previously described for oviedomycin BGC activation [42]. Longer Aur1O homologues (Aur1O, FarD, ChaP, SaqP, PgaK, SchP11, Med-ORF27, GcnN), which contain specific conserved C-terminal domain VI (Figure 7a), may similarly interact with their cognate DNA-binding activators (as in the case of Aur1O and Aur1P interaction) to activate the expression of biosynthetic genes in their respective BGCs. These transcriptional activators belong mainly to the ARR family (Appendix A).

In conclusion, a new family of transcriptional coactivators was identified. This new family is divided into two distinct subclasses. The shorter coactivators contain only the first five domains (I, II, III, IV, V) (Figure 7a) and can interact with the short OvmW homologue [42]. The longer coactivators contain six conserved domains (I, II, III, IV, V, VI) (Figure 7a) and can interact with transcriptional activators mainly from the ARR family. This type of regulation of antibiotic BGCs is novel. Although many families of transcriptional regulators regulating antibiotic production have been characterized [1,5,6], such coactivation of a DNA binding transcriptional activator with some other associated protein has not been described, with the exception of Aur1O and its two homologues, TylU and OvmZ. Moreover, this type of regulation is likely to be widespread because Blast search in databases revealed homologous *aur1O* genes in many *Streptomyces* species sequenced, most of which are found in unknown secondary metabolite BGCs. Examination of the 99 most similar Aur1O homologues (Appendix A) revealed that almost all of their genes were located in BGCs (one corresponding gene was not close to any BGC and the three corresponding nucleotide sequence contigs were very short), in most cases in close proximity to the gene encoding the putative ARR activator, as in the cases of *aur1O*/*aur1P*. Almost all BGCs belong to angucycline type II PKSs (80 cases) or other type II PKSs (14 cases). However, one BGC belongs to the non-ribosomal peptide synthases (NRPS). Therefore, although rarely, this system may be involved in the regulation of some other BGC classes. 

In future, it would be interesting to study this interesting mechanism of this activation in both branches. Structural analysis of both protein pairs would help to elucidate this mechanism and domain interactions. This is a challenge not only for our group, but also other partners who are investigating similar BGCs. 

## 3. Material and Methods

### 3.1. Bacterial Strains, Plasmids, and Culture Conditions

*S. lavendulae* subsp. *lavendulae* CCM 3239 wild type strain [11] was obtained from the Czech Collection of Microorganisms (Brno, Czech Republic). For sporulation, the strain was grown on solid Bennet medium as described in [15]. For the detection of auricin, *S. lavendulae* subsp. *lavendulae* CCM 3239 wild type and ∆*aur1O* mutant strains were cultivated in liquid Bennet medium to various growth phases as previously described [15]. For RNA isolation, 5 × 10^8^ colony-forming units (CFU) of *S. lavendulae* subsp. *lavendulae* CCM 3239 wild type and ∆*aur1O* mutant spores were inoculated into 50 mL Bennet medium in 250 mL Erlenmeyer flasks and the culture was grown on an orbital shaker at 270 r.p.m. and 28 °C to different growth stages. *Escherichia coli* DH5a (Invitrogen, Waltham, MA, USA) was used as a host for standard cloning experiments. *E. coli* BW25113/pIJ790 was used as a host for PCR-targeted gene disruption using the AprR plasmid pIJ773, and *E. coli* ET12567/pUZ8002 was used as a non-methylating host [21]. The *Streptomyces* integrative plasmid pAPHII15 [44] was used for the complementation studies. The luciferase reporter plasmid pMU1s*containing the synthetic *luxCDABE* operon was used for detection of promoters in *Streptomyces* [26]. The conditions for *E. coli* growth and transformation were described in [22]. Luria–Bertani (LB) medium was used for *E. coli* growth. If required, the media were supplemented with 100 μg/mL Amp (Sigma-Aldrich, Darmstadt, Germany), 50 μg/mL Apr (Sigma-Aldrich, Darmstadt, Germany), 50 μg/mL Kan (Sigma-Aldrich, Darmstadt, Germany), 40 μg/mL chloramphenicol (Sigma-Aldrich, Darmstadt, Germany), 50 μg/mL streptomycin (Sigma-Aldrich, Darmstadt, Germany). 

### 3.2. Recombinant DNA Techniques 

Standard DNA manipulation methods were performed as described in [22]. Chromosomal DNA from *S. lavendulae* subsp. *lavendulae* CCM 3239 wild type and ∆*aur1O* mutant strains was prepared according to [45]. Southern blot hybridization analysis was performed as described in [22]. 1 μg of DNA was digested with restriction enzymes, separated by electrophoresis in a 0.8% (*w*/*v*) agarose gel, and transferred to a Hybond N membrane (Roche, Mannheim, Germany). Hybridization was performed according to the standard DIG protocol (Roche, Mannheim, Germany) using a DIG-labelled probe (850-bp DNA fragment covering the *aur1A* gene, Figure 1a) prepared by PCR amplification using primers aur1AFw and aur1ARv (Appendix A). Signals were detected by DIG chemiluminescent detection kit using CSPD (Roche, Mannheim, Germany). 

### 3.3. Disruption of the S. lavendulae subsp. lavendulae CCM 3239 aur1O Gene 

The PCR targeted REDIRECT procedure [21] was used to delete the entire coding region of the *aur1O* gene in *S. lavendulae* subsp. *lavendulae* CCM 3239. The AprR cassette from template plasmid pIJ773 was PCR amplified using primers Aur1OdDir and Aur1OdRev (Appendix A). The resulting PCR product was used to electroporate *E. coli* BW25113/pIJ790 containing the cosmid pCosSA25 [17]. The correct replacement of the *aur1O* gene in the resulting cosmid pCosSA25-aur1O was verified by restriction mapping. The recombinant cosmid was transformed into the non-methylating *E. coli* ET12567/pUZ8002 strain and introduced into *S. lavendulae* subsp. *lavendulae* CCM 3239 by conjugation. Colonies were screened for AprR and Kan sensitivity, indicating a double crossover. Four such colonies were identified, resulting in mutant strains *S. lavendulae* ∆*aur1O::AprR1, 2, 3, 4*, and the correct double-cross event was confirmed by Southern-blot hybridization. All four clones had similar phenotypes. One representative strain was selected for further study. Plasmid pAPHII15-aur1O used to complement the *aur1O* mutation was prepared by inserting a PCR-amplified DNA fragment containing the entire *aur1O* gene, including its RBS site, into the integrative *Streptomyces* expression plasmid pAPHII15 [44]. A 900-bp *Cla*I-*Xba*I DNA fragment was prepared by PCR amplification using chromosomal DNA from *S. lavendulae* subsp. *lavendulae* CCM 3239 as a template and the primers aur1OCla and aur1OXba (Appendix A) to introduce a *Cla*I site upstream and an *Xba*I site downstream of the *aur1O* gene. *Pfu* DNA polymerase (Stratagene, La Jolla, CA, USA) was used to ensure a high fidelity of DNA synthesis during PCR. The DNA fragment was digested with *Cla*I and *Xba*I and cloned into pAPHII15 digested with the same enzymes, resulting in pAPHII15-aur1O, which was verified by nucleotide sequencing. 

### 3.4. RNA Isolation and S1-Nuclease Mapping

Isolation of total RNA from *S. lavendulae* subsp. *lavendulae* CCM 3239 and high-resolution S1 nuclease mapping were performed according to [46]. RNA samples (40 μg) were hybridized to approximately 0.02 pmol DNA probe labelled at one 5′ end with [γ-^32^P] ATP (producing approximately 10^6^ dpm/pmol of probe) and treated with 120 U of S1-nuclease (New England Biolabs, San Diego, CA, USA). The DNA probe for the *aur1Ap* promoter (274 bp DNA fragment) was prepared by PCR amplification using chromosomal DNA from *S. lavendulae* subsp. *lavendulae* CCM 3239 as a template and the 5′ end-labelled primer aur1AS1Rev from the *aur1A* coding region and the unlabeled primer aur1AS1Dir from the *aur1A* upstream region. The control *hrdBp2* promoter DNA probe was described in [24]. Oligonucleotides were labelled at the 5′ end with [γ-^32^P] ATP (ICN, Costa Mesa, CA, USA, 4500 Ci/mmol) and T4 polynucleotide kinase (New England Biolabs, San Diego, CA, USA) as described in [22]. The protected DNA fragments were analyzed on DNA sequencing gels (6% polyacrylamide containing 8 M urea) along with G+A and T+C sequencing ladders derived from the end-labeled fragments [25]. 

### 3.5. Construction of luxCDABE-Based Luciferase Reporter Plasmids and Bioluminescence Measurement

The luciferase reporter plasmid pMU1s* containing the synthetic *luxCDABE* operon [26] was used to detect promoter activity. A 189-bp DNA fragment containing the *aur1Ap* promoter, including its Aur1P-binding site [16], was PCR amplified with high fidelity *Pfu* DNA polymerase (Stratagene, La Jolla, CA, USA), *S. lavendulae* subsp. *lavendulae* CCM 3239 chromosomal DNA as a template and the primers aur1ApDir and aur1ApRev (Appendix A). The fragment was digested with *Bam*HI and *Kpn*I and cloned into pMU1s* digested by the same restriction enzymes, resulting in pMU1s-Ap1. The *aur1P* gene, including its *aur1Pp* promoter [16], was similarly PCR amplified using the primers aur1PpDir and aur1PpRev (Appendix A). The resulting 900-bp DNA fragment was digested with *Eco*RV and *Bst*BI and cloned into pMU1s-Ap1 digested with the same restriction enzymes, resulting in pMU1s-PAp1. The *aur1P-aur1O* operon, including the *aur1Pp* promoter, was similarly PCR amplified using the primers aur1PpDir and aur1OpRev (Appendix A). The resulting 1900-bp DNA fragment was digested with *Eco*RV and *Bst*BI and cloned into pMU1s-Ap1 digested with the same restriction enzymes, resulting in pMU1s-POAp1. The nucleotide sequence of all the recombinant plasmids was checked by sequencing using primers luxCrev and pMU1fd (Appendix A). The resulting recombinant plasmids were conjugated to the heterologous *S. coelicolor* M1146 host strain [27], and eight independent AprR clones from each construct were selected to determine luminescence. Spores from each clone, approximately 10^5^ CFU, were inoculated into each well of white 96-well plates (Sigma-Aldrich, Darmstadt, Germany) containing 0.25 mL of solid Bennet medium and grown to a confluent lawn. The plates were incubated at 28 °C and luminescence was measured every 4 h in a Synergy HT microplate reader (Bio-Tek Instruments, Winooski, VT, USA) in RLU. The background luminescence of an empty well without inoculated strain was approximately 200 RLU. 

### 3.6. BACTH System to Investigate Protein-Protein Interactions 

A well-established BACTH system, which is based on functional complementation of the T18 and T25 adenylate cyclase fragments of *Bordetella pertussis* in a *cya*-deficient *E. coli* BTH101 strain [28], was used to assess the potential interaction between Aur1P and Aur1O proteins. The complete *aur1P* and *aur1O* genes were amplified by PCR with high fidelity *Pfu* DNA polymerase (Stratagene, La Jolla, CA, USA), *S. lavendulae* subsp. *lavendulae* CCM 3239 chromosomal DNA as a template and selected primes (Appendix A) to introduce *Xba*I and *Nde*I sites next to the translation initiation codon, and *Xho*I and *Kpn*I sites downstream of the stop codon. The amplified DNA fragments were digested with *Xba*I and *Kpn*I and then ligated into the plasmid pKT25 or pUT18C digested by the same restriction enzymes, resulting in the final recombinant plasmids. The *aur1P* gene was amplified with the primers aur1PDHdir and aur1PDHrev (Appendix A), and a 771-bp DNA fragment was cloned into pUT18C or pKT25, resulting in pUT18C-aur1P or pKT25-aur1P. The *aur1O* gene was amplified with the primers aur1ODHdir and aur1ODHrev (Appendix A), and an 882-bp DNA fragment was cloned into pUT18C or pKT25, resulting in pUT18C-aur1O or pKT25-aur1O. The nucleotide sequence of all the genes was checked by sequencing using primers KT25dir and KT25rev or UT18Cdir and UT18Crev (Appendix A). Combinations of the resulting pUT18C and pKT25 constructs were co-transformed into a *cya*-deficient *E. coli* BTH101 strain and screened on LB medium supplemented with 0.5 mM IPTG (Sigma-Aldrich, Darmstadt, Germany), 40 μg/mL X-Gal (Sigma-Aldrich, Darmstadt, Germany). Amp and Kan were added to select the plasmids and streptomycin to select the strain. Colonies were screened after two days at 30 °C. Quantitative measurements of β-galactosidase activities in sets of three independent transformants for each plasmid combination were performed according to [47]. The statistical significance of datasets was analyzed by two-tailed Student’s *t* test with *p* < 0.05 considered as significant. 

### 3.7. Analysis of Auricin Production

Spores (4 × 10^9^ CFU) of *S. lavendulae* subsp. *lavendulae* CCM 3239 wild type and ∆*aur1O* mutant strains were inoculated into 50 mL Bennet medium in 250 mL Erlenmeyer flasks and cultures were grown on an orbital shaker at 270 r.p.m. and 28 °C. 4 mL aliquots were taken from the cultures at various time points and centrifuged at 6000× *g* for 5 min. The supernatants were extracted twice with an equal volume of ethyl acetate (Merck, Darmstadt, Germany), the residual water was removed with sodium sulfate (Sigma-Aldrich, Darmstadt, Germany) and sodium chloride (Sigma-Aldrich, Darmstadt, Germany), and the extracts were evaporated in vacuo. Such samples were dissolved in 100 μL of 96% (*v*/*v*) ethanol (Merck, Darmstadt, Germany) and analyzed by TLC and HPLC as described previously [14]. Briefly, 10 μL aliquots were subjected to TLC analysis on silica gel 60 F_254_ plates (Merck, Darmstadt, Germany) with *n*-butanol saturated with water. Dried TLC plates were analyzed by biochromatography, covered with soft nutrient agar [48] containing fresh *B. subtilis* culture. The plates were incubated at 37 °C for 16 h and visually screened for growth inhibitory zones. HPLC analysis was performed using a Shimadzu LC-10A HPLC system with a SPD-20A UV-VIS detector (Shimadzu, Kyoto, Japan) and detection at 254 nm. Samples of 10 μL were applied to an OmmiSpher 5 C_18_ column (5 μm, 250 × 4.6 mm, Varian, Lake Forest, CA, USA) and eluted with a linear gradient of acetonitrile (solvent B) and 0.5 % (*v*/*v*) acetic acid in water (solvent A) at a flow rate of 1 mL/min. A 30-min linear gradient from 20% to 100% solvent B was used. Auricin showed a retention time of 8.7 min under these conditions and was confirmed by ESI-MS analysis as a molecular ion [M + H] of *m*/*z* = 542.2 with positive mode detection [14]. The ESI spectrum was measured using a Velos Pro Hybrid Ion Trap-Orbitrap Mass Spectrophotometer (Thermo Scientific, Waltham, MA, USA).

## Figures and Tables

**Figure 1 ijms-23-02455-f001:**
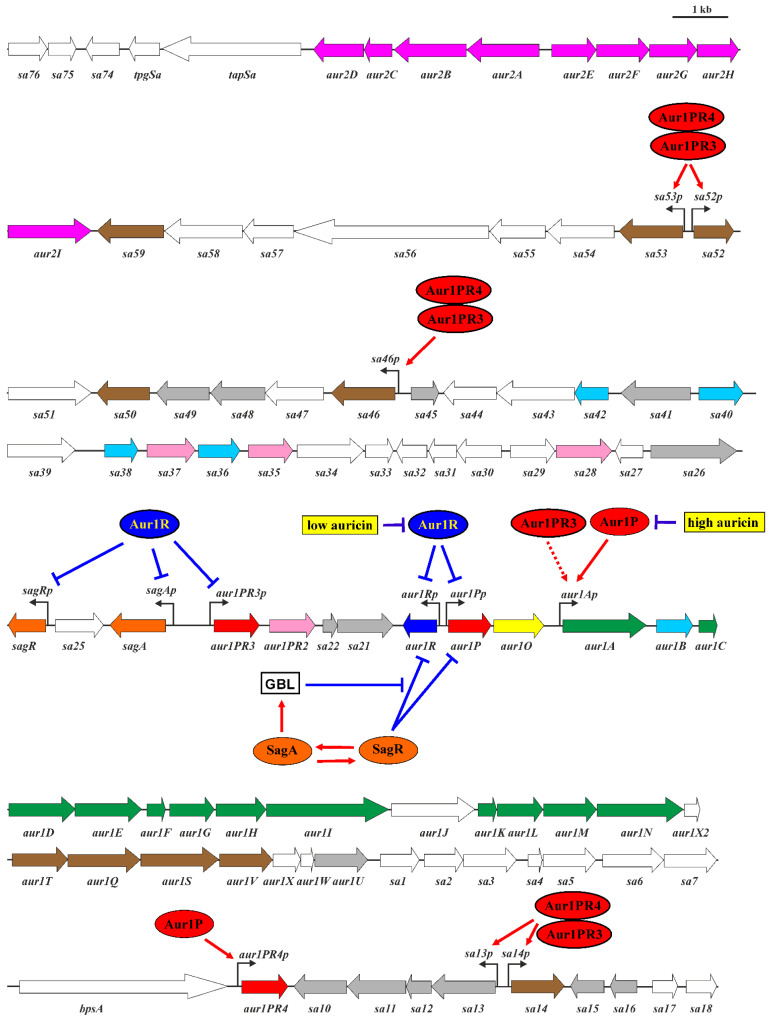
Auricin BGC and the model of the regulation of auricin biosynthesis [8,12,20]. Details of individual genes and their products are provided in GenBank Acc. No. KJ396772. The green arrows correspond to the auricin aglycone biosynthetic genes; the grey arrows to putative polyketide biosynthetic genes; the brown arrows to D-forosamine biosynthetic and attachment genes; the yellow arrow to the *aur1O* gene; the red arrows to the positive regulatory genes, with red-dark color having a role in regulation of auricin biosynthesis and red-light color having no role in regulation; the blue arrows to the negative regulatory genes, with blue-dark color having a role in regulation of auricin biosynthesis and blue-light color having no role in regulation; the orange arrows to the GBL autoregulator-receptor system genes; the purple arrows to the other type II PKS BGC. Bent arrows indicate the position and direction of promoters. Red lines ending with an arrow indicate transcriptional activation, whereas blue lines ending with a perpendicular line indicate repression. Indirect activation is indicated by a broken arrow.

**Figure 2 ijms-23-02455-f002:**
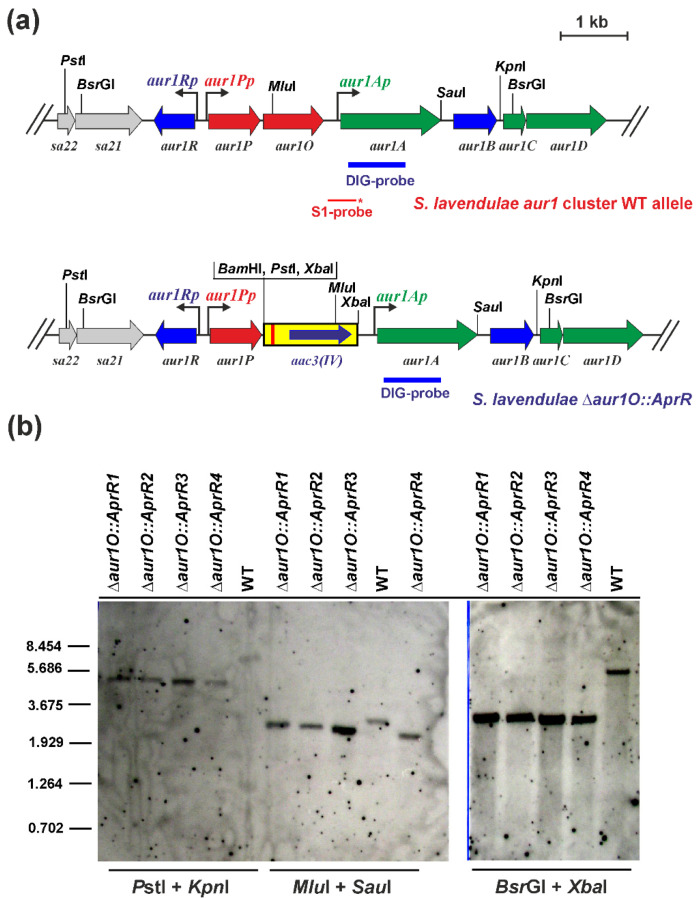
(**a**) Physical map of the *aur1* BGC around the *aur1O* gene in wild-type *S. lavendulae* subsp. *lavendulae* CCM 3239 [8] and a disrupted *aur1O* gene allele. Genes are indicated by arrows. The green arrows correspond to the auricin biosynthetic genes, the red arrows to the positive regulatory genes, and the blue-arrows to the negative regulatory genes. The yellow box indicates the AprR *aac3*(*IV*) gene with *oriT* origin of transfer from pIJ773 [21]. The bent arrows indicate the positions and directions of transcription from the indicated promoters. The thin red line below the map represents the DNA fragment used as a probe in S1 nuclease mapping (the 5′-labelled end is marked with an asterisk). The blue bar below the maps represents the probe used for Southern hybridization analysis. Relevant restriction sites are indicated. (**b**) Southern blot hybridization analysis of chromosomal DNA from four independently obtained *S. lavendulae* ∆*aur1O::AprR* clones and wild-type *S. lavendulae* subsp. *lavendulae* CCM 3239 (WT) as reference. 1 μg of DNA from the corresponding strain was digested with the indicated restriction endonucleases, separated by electrophoresis in a 0.8% (*w*/*v*) agarose gel, and transferred to a Hybond N membrane (Roche) as described in [22]. Hybridization was performed according to the standard DIG protocol (Roche, Germany) using a DIG-labeled probe covering the *aur1A* gene. Lambda DNA-*Bst*EII digest was used as a size standard.

**Figure 3 ijms-23-02455-f003:**
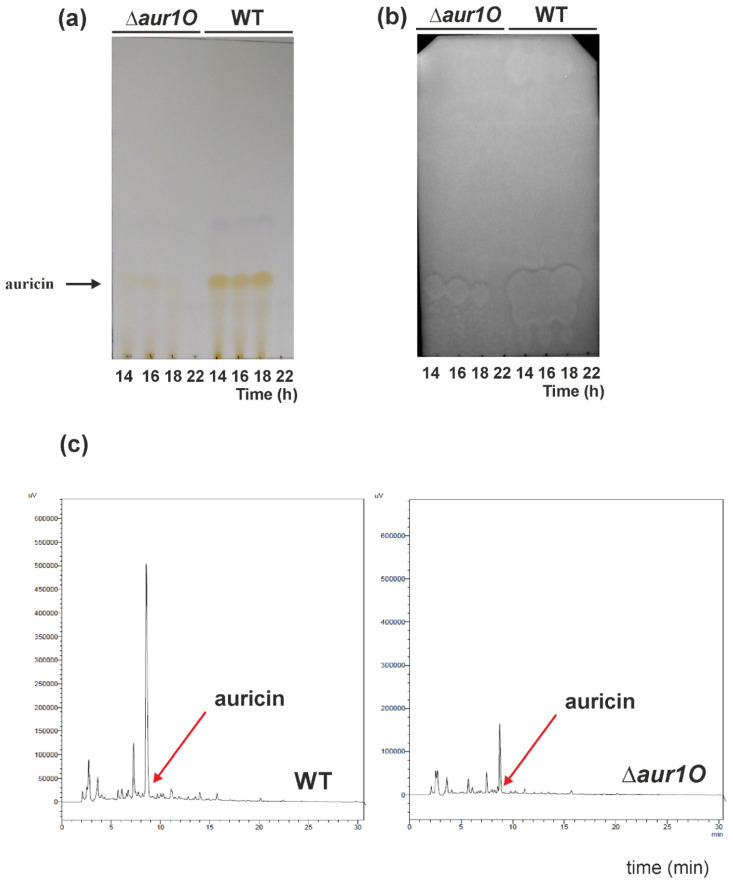
Analysis of auricin production in wild-type (WT) and *aur1O* mutant by TLC (**a**) followed by biochromatography with *B. subtilis* test strain. (**b**) 10 μL of ethyl acetate extracts from each strain grown in Bennet medium to indicated time points were separated by TLC. For TLC biochromatography, 1 μL aliquots of the ethyl acetate extracts were separated by TLC and overlaid with *B. subtilis* as described in Material and Methods. Arrows indicate the yellow spot and the inhibition zone corresponding to auricin [14]. (**c**) HPLC analysis of auricin production by wild-type *S. lavendulae* subsp. *Lavendulae* CCM 3239 (WT) and *aur1O* mutant (∆*aur1O*) strains. 10 μL of the ethyl acetate extracts from each strain grown 14 h in Bennet medium were analyzed (details described in Material and Methods). The arrow indicates the position of the auricin peak.

**Figure 4 ijms-23-02455-f004:**
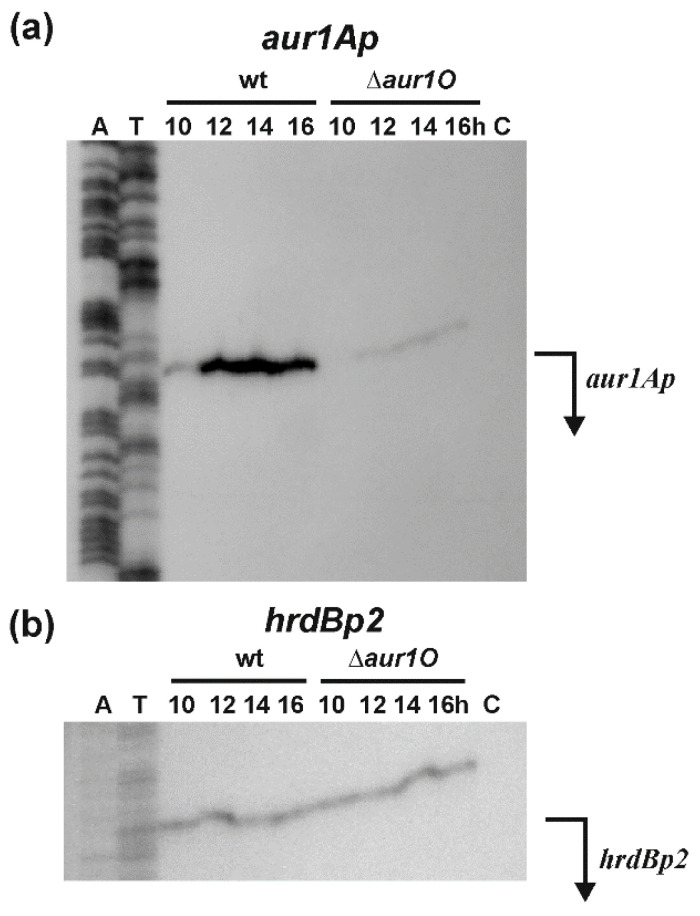
(**a**) High-resolution S1-nuclease mapping of the transcription start site (TSS) for the *aur1Ap* promoter in wild-type *S. lavendulae* subsp. *lavendulae* CCM 3239 (WT) and *aur1O* mutant strains (∆*aur1O*). The 5′-labelled DNA fragment (Figure 2a) was hybridized to RNA isolated from cultures grown in liquid Bennet medium at the indicated time points (corresponding to different growth phases). *E. coli* tRNA was used as a control (lane C). (**b**) Control S1-nuclease mapping with the same RNA samples and a DNA probe for the *hrdBp2* promoter [24]. The RNA-protected DNA fragments were analyzed on DNA sequencing gels together with G+A (lane A) and T+C (lane T) sequencing ladders derived from the end-labelled fragments [25]. Thick bent horizontal arrows indicate the positions of RNA-protected fragments corresponding to TSS of the promoters.

**Figure 5 ijms-23-02455-f005:**
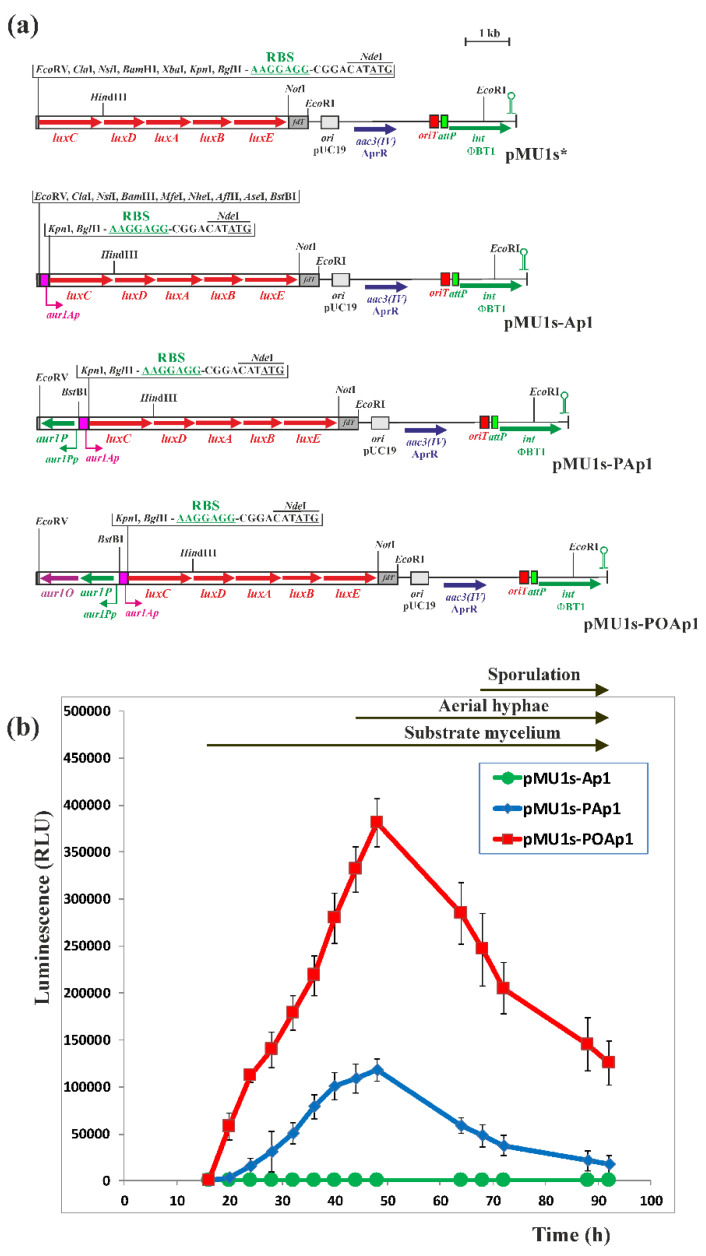
Activation of the *aur1Ap* biosynthetic promoter by Aur1P and Aur1O in the heterologous *S. coelicolor* M1146 system. (**a**) Maps of the *luxCDABE* reporter plasmid pMU1s* [26] and recombinant plasmids containing the *aur1Ap* promoter with and without the *aur1P* and *aur1O* genes. Arrows indicate corresponding genes and bent arrows promoters. The plasmids contain the synthetic operon *luxCDABE*, the *aac3*(*IV*) AprR gene, the *oriT* transfer origin, the *E. coli* ColE1 replication origin from pUC19, phage *fd*, major transcription terminator, and *attP* attachment site and ΦBT1 phage integrase gene. Relevant restriction sites are indicated. (**b**) The activity of the *luxCDABE* operon after fusion with the corresponding promoter DNA fragments. The corresponding plasmids were conjugated to *S. coelicolor* M1146 and luminescence was measured in a Synergy HT microplate reader in RLU after growth and differentiation on solid Bennet medium in 96-well plates at the indicated time points. Each point represents the mean of eight assays, and error bar indicates the standard deviation from the mean. The arrows above the graph indicate the positions of developmental stages.

**Figure 6 ijms-23-02455-f006:**
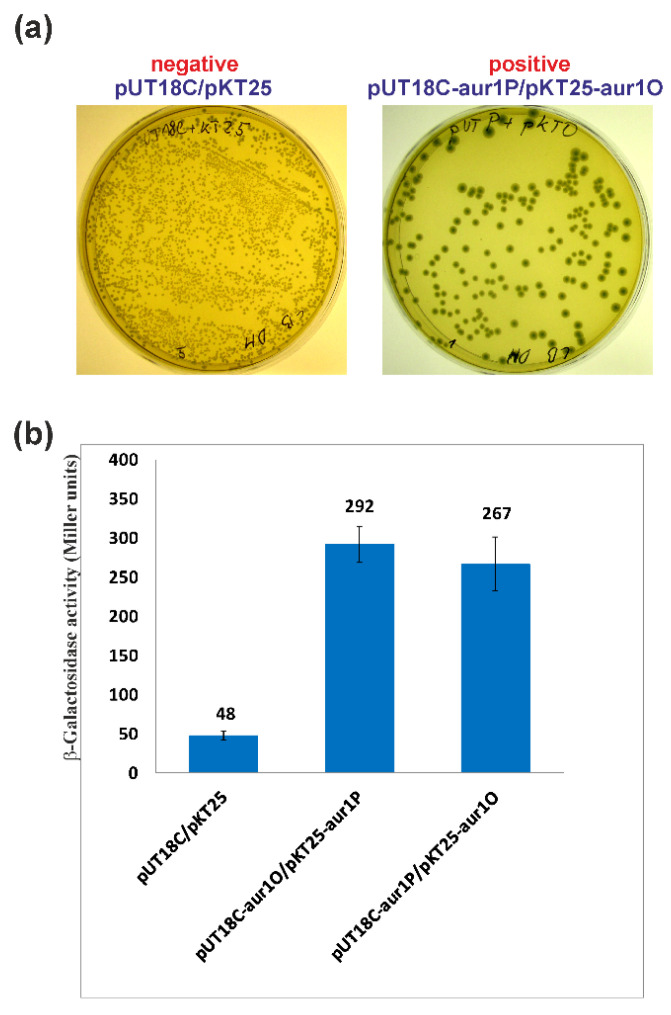
Analysis of the Aur1O-Aur1P interaction using the BACTH system [28]. Combinations of the plasmids were transformed into *E. coli* BTH101 and screened on solid LB medium supplemented with ampicillin (Amp), kanamycin (Kan), IPTG, and X-Gal. (**a**) Example of plates with negative and positive combinations of plasmids. (**b**) β-galactosidase activity for each plasmid combination determined in triplicates as described in Materials and Methods. The error bars indicate standard deviations from the mean (given by the values above each bar).

**Figure 7 ijms-23-02455-f007:**
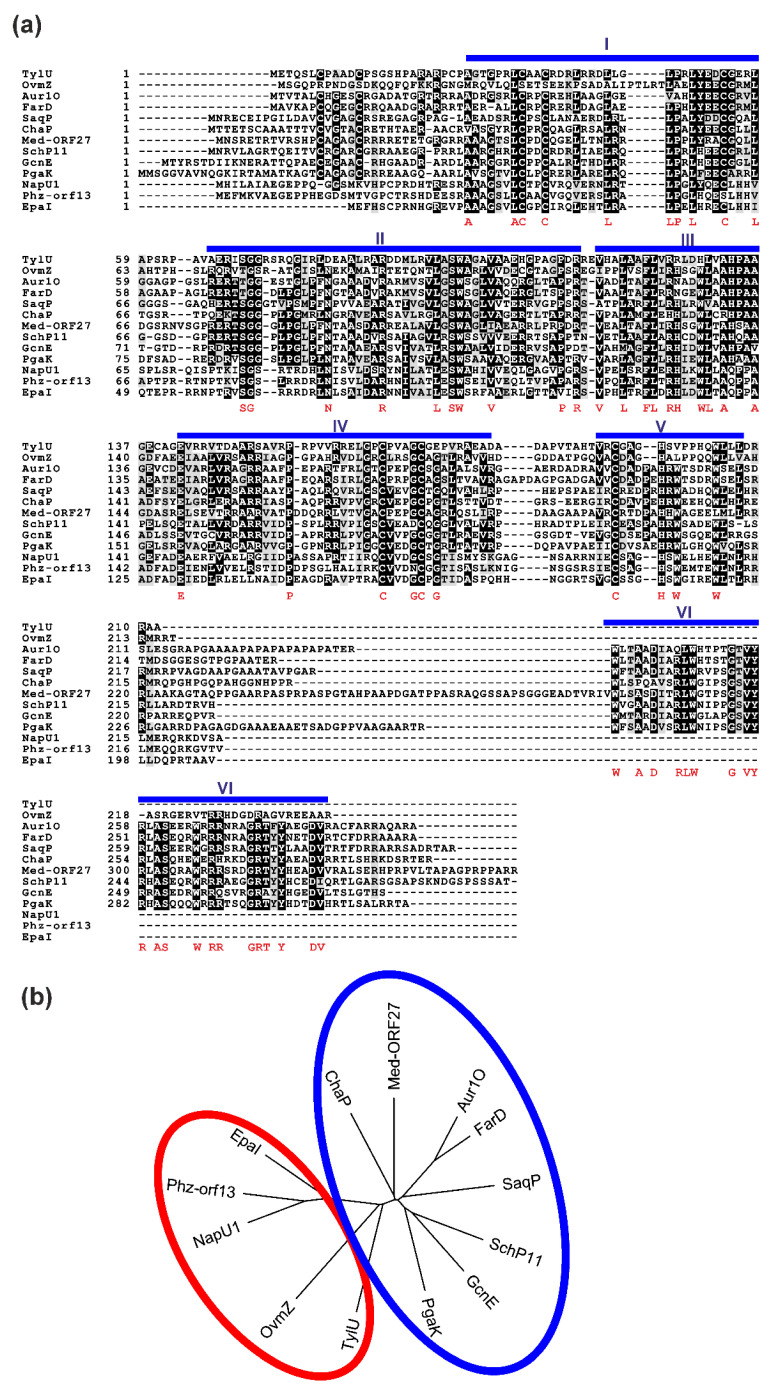
(**a**) A comparison of Aur1O with similar proteins from BGCs. Protein sequences (and accession numbers) are as follows: Aur1O of *S. lavendulae* subsp. *lavendulae* CCM 3239 (AAK59995), TylU of *S. fadiae* T59235 tylosin BGC (AAD40806), OvmZ of *S. antibioticus* ATCC 11891 oviedomycin BGC (CAG14961), FarD of *S. lavendulae* FRI-5 *lac* BGC for unknown aromatic polyketide (BAG74709), SaqP of *Micromonospora* sp. Tu6368 saquayamycin BGC (ACP19350), ChaP of *S. chattanoogensis* L10 chattamycin BGC (AIU99193), Med-ORF27 of *Streptomyces* sp. AM-7161 medermycin BGC (BAC79023), SchP11 of *Streptomyces* sp. SCC-2136 *sch* BGC for angucyclines Sch 47554 and Sch 47555 (CAH10120), GcnE of *S. lusitanus* SCSIO LR32 grincamycin BGC (AGO50608), PgaK of *Streptomyces* sp. PGA64 gaudimycin BGC (AHW57770), NapU1 of *Streptomyces* sp. CNQ-525 napyradiomycin BGC (ABS50476), Phz-orf13 of *S. tendae* Tue1028 phenazine BGC (AFS18594), EpaI of *Kitasatospora* sp. HKI 714 endophenazine BGC (AHW81468). The conserved domains are indicated by blue bars above the sequences. Identical residues are highlighted in black. Similar residues are shaded. The highly conserved residues are listed below the sequences in red. The numbers refer to the deposited amino acid sequences in databases. (**b**) Phylogenetic tree of these Aur1O homologues; the separate branches are indicated by circles.

**Figure 8 ijms-23-02455-f008:**
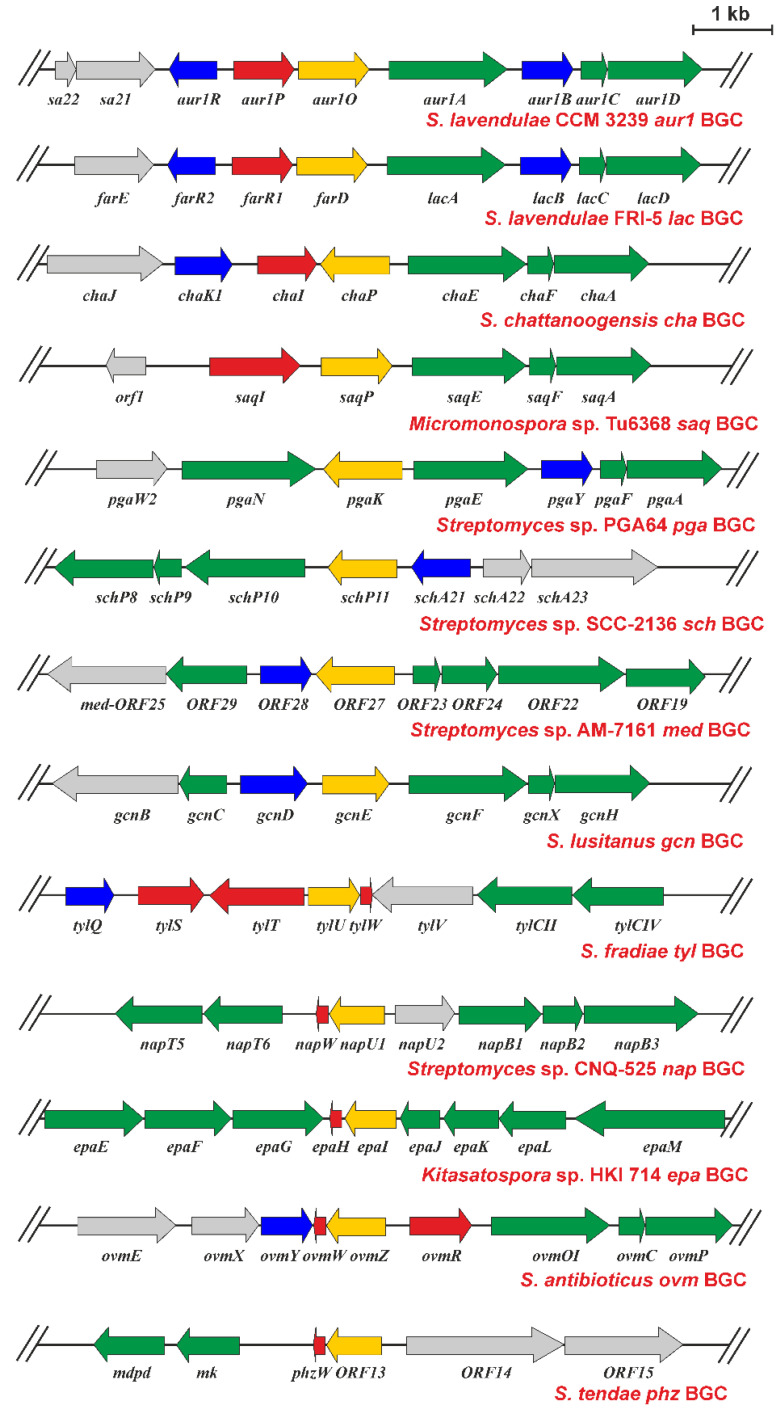
Physical maps of *S. lavendulae* subsp. *lavendulae* CCM 3239 *aur1* BGC around the *aur1O* gene [8] and other BGCs containing the *aur1O* homologue. Genes are indicated by arrows. The green arrows correspond to the biosynthetic genes, the yellow arrows to the *aur1O* homologue, the red arrows to the positive regulatory genes, and the blue arrows to the negative regulatory genes. BGC accession numbers are: *S. lavendulae* FRI-5 *lac* BGC for unknown aromatic polyketide (AB434932, LC209815), *S. chattanoogensis* L10 chattamycin (*cha*) BGC (KM264312), *Micromonospora* sp. Tu6368 saquayamycin (*saq*) BGC (FJ670504), *Streptomyces* sp. PGA64 gaudimycin (*pga*) BGC (AY034378), *Streptomyces* sp. SCC-2136 angucyclines Sch 47554 and Sch 47555 (*sch*) BGC (AJ628018), *Streptomyces* sp. AM-7161 medermycin (*med*) BGC (AB103463), *S. lusitanus* SCSIO LR32 grincamycin (*gcn*) BGC (KC962511), *S. fradiae* T59235 tylosin (*tyl*) BGC (AF145049), *Streptomyces* sp. CNQ-525 napyradiomycin (*nap*) BGC (EF397639), *Kitasatospora* sp. HKI 714 endophenazine (*epa*) BGC (KJ207079), *S. antibioticus* ATCC 11891 oviedomycin (*ovm*) BGC (AJ632203), *S. tendae* Tue1028 phenazine (*phz*) BGC (JQ659263).

## Data Availability

Data are available on request.

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
