# Peer review of "A New Family of Transcriptional Regulators Activating Biosynthetic Gene Clusters for Secondary Metabolites"

_ijms, 2022, doi:10.3390/ijms23052455_

Round 1
Reviewer 1 Report
In the paper entitled: A new family of transcriptional regulators activating biosynthetic gene clusters for secondary metabolites, the authors describe a new type of regulatory protein involved in polyketide cluster activation. The experimental part is done properly using appropriate techniques and characterizes the participation of Aur1O protein in the activation pathway of auricine synthesis cluster.
The presented manuscript should be substantially rewritten and completed before publication.
The major issues are:
- Authors propose to create a new family of activators based on homology/similarity to the Aur1O protein. However, they do not provide clear criteria or features to characterize these proteins. The text needs to be amended and edited so that readers can use this information when analyzing other regulatory proteins.
- Readers can get the impression that the manuscript was written some time ago. Especially in the introduction section the authors refer to quite old review papers. This part should be rewritten. Especially because the BGC described here contains a butanolide system which, as more recent reports indicate, may interact with other elements of secondary metabolism in a given organism via regulators described as cluster-specific.
- In the experiment of aur1Ap promoter activation in a heterologous system, it is not clear how the aur1Pp promoter is activated (activation of aur1p and aur1O proteins). Does this promoter participate in some specific activation pathway, or is it constitutively active in S. coelicolor? The authors should note that S. coelicolor strain M1146 has a butanolide system left active. Although it is labeled as deltacpk the cpk cluster is only deleted in the SCO6270 -SCO6287 region, and both the butanolide system and the SCO6288 regulator remain and may be active. The authors should clarify this situation.
Minor issues:
- It seems unnecessary to describe the types of polyketide synthases in the introduction. Please consider shortening or removing this paragraph.
- In the caption for figure 5, is the use of the name M1145 correct ?(line 288) N
- In figure 3 there should be a separate scale for the graph derived from the mutant.
Author Response
All details are in the uploaded file responses_to_reviewer1.pdf

Reviewer 2 Report
Evaluation of the article “A new family of transcriptional regulators activating biosynthetic gene clusters for secondary metabolites”, by Novakova et al
General comments
The article by Novakova et al. is an interesting study of the protein encoded by aur1O gene of the auricin gene cluster of Streptomyces lavendulae (formerly identified as S. aureofaciens). This gene encodes a protein without clear similarity to other regulatory protein involved in regulation of gene clusters in Streptomyces. Using several classical and molecular genetic tools the authors identify that this gene encodes a protein that works as a coactivator of the transcriptional activator Aur1P, previously characterized by the same group. This group has reported four transcriptional regulators of the auricin gene cluster. Full characterization of the regulatory cascade that controls auricin biosynthesis is important. The experimental techniques used are correct and the authors find a mechanism of coactivation that is relevant for additional characterization of other antibiotic gene clusters. However, there are some obscure points that are indicated below and should be answered by the authors
Specific comments
- Lines 74-75. The authors describe: “The transcriptional activator StrR, which controls the biosynthesis of aminoglycoside antibiotic streptomycin in S. griseus, is a prototype of another StrR family“. This sentence has no clear meaning since it repeats twice the Str word.
- Lines 77 “StrT…belongs to the ParB-Spo0J family of DNA segregation proteins”. Please provide a reference since all other regulators have specific references, except this one.
- Line 142 says “…Represent a new family of transcriptional activators involved in the regulation of secondary metabolite biosynthesis…”. This designation is confussing. They should be named throughout the entire text as a co-activator protein since it does not bind directly to the DNA.
- Results and Discussion. The authors constructed a mutant by replacing the entire aur10 by the complete apramycin resistance gene. There is a possible problem with this replacement since if the complete apramycin resistance gene is expressed it may affect the growth or the auricin gene cluster expression and the antibiotic production. An additional control should be included, i.e. a strain in which the apramycin resistance gene is eliminated or changed for a small DNA fragment.
- Line 265. It says “The activity of the aur1Ap promoter in this construct increased during growth and its maximum (on average 118,030 RLU) coincided with the onset of aerial mycelium formation (Figure 5b)”, and in the abstract “Auricin is produced in a narrow interval of the growth phase…”. The authors do not provide in the text any information about how the time course affect expression of the operon. Does the pH affect expression of the genes?
- Lines 293-294 the authors state “Previous results have suggested that Aur1O may exert its activation function on the aur1Ap promoter through interaction with Aur1P”, but they do not provide any evidence or reference about those previous studies.
- In the last section the authors make bioinformatic searches of Aur1O homologues in other BGCs. In summary, they find proteins similar in other twelve gene clusters; these proteins are grouped in those of type II PKS for angucycline aromatic polyketides, and another that may be related to macrolides such as tylosin. But in line 413 the authors state “Moreover, this type of regulation is likely to be widespread because Blast search in databases revealed homologous aur1O genes in many Streptomyces species sequenced, most of which are found in unknown secondary metabolite BGCs”. The authors do not explain whether the co-activator gene is present in BGCs for non-ribosomal peptide derived metabolites, for aminoglycosides or for terpenoids, so the authors should comment on the limitation or extension of this mechanism, since the co-regulation requires one gene for the transcriptional regulator and other for the co-activation that must interact between them.
Author Response
All details are in the uploaded responses_to_reviewer2.pdf. Please see the attachment.
